# Varying isoleucine level to determine effects on performance, egg quality, serum biochemistry, and ileal protein digestibility in diets of young laying hens

S. Ullah[1], Y. A. Ditta[1]*, A. J. King[2]*, T. N. Pasha[1], A Mahmud[3], K. A. Majeed[4]

1 Department of Animal Nutrition, University of Veterinary and Animal Sciences, Lahore, Pakistan,
2 Department of Animal Science, University of California, Davis, Davis, CA, United States of America,
3 Department of Poultry Production, University of Veterinary and Animal Sciences, Lahore, Pakistan,
4 Department of Physiology, University of Veterinary and Animal Sciences, Lahore, Pakistan

* yasirad787@gmail.com (YAD); ajking@ucdavis.edu (AJK)

## Abstract

To ascertain an appropriate level of isoleucine for LSL-LITE layers (23- to 30-week-old), diets containing total isoleucine concentrations (levels) of 0.66 (Control), 0.69, 0.72, 0.75, 0.78, 0.81, and 0.84% were fed as 7 treatments (2730 kcal/kg metabolizable energy) x 7 replicates x 10 birds per replicate. Significance for performance, egg quality, serum biochemistry, and ileal digestibility of protein was determined at $P \leq 0.05$. Level, week, and level*week (L*W) were significant for production, egg mass, and feed intake. Level and week were significant for FCR. Week was significant for weight gain. Level was significant for egg weight, specific gravity, and shell thickness; week was also significant for these external egg parameters as well as shape index and proportional shell thickness. L*W was significant for all except shape index. For internal egg measurements, level was significant for proportional yolk, proportional albumen, yolk index, and yolk:albumen. Week was significant for internal egg parameters while L*W significantly affected Haugh unit, proportional albumen weight, yolk index, albumen index, and yolk color. Level was significant for globulin and glucose in serum. Isoleucine at 0.72%, 0.81%, and 0.84% produced the lowest FCR, an important standard in the poultry industry. Considering the low FCR of 1.45 and cost for inclusion as a dietary ingredient, 0.72% isoleucine was chosen for further studies with varying quantities of other branched chain amino acids in diets for young laying hens.

## Introduction

It has been acknowledged that earlier recommendations for dietary amino acids for poultry may not be adequate [1]. Since 1994, researchers have continued to recommend appropriate levels of amino acids in layer diets. For instance, recommended optimum levels of isoleucine have included 550 to 660 mg/hen/day [2–4].

Huyghebaert et al. determined that the daily isoleucine requirements of layers (26- to 36-week-old) did not decrease while egg output increased [5]. On the other hand, several investigators noted that age and genetic variety (breed) of layers prompted further study

are available at DOI https://doi.org/10.5061/dryad.m905qfv1t.

**Funding:** The funders had no role in study design, data, collection, decision to publish, or preparation of the manuscript.

**Competing interests:** I have read the journal's policy and the authors of this manuscript have no competing interests.

associated with amino acids [6–8]. Values from producers of five layer breeders ranged from 640 to 767 mg/bird/day for isoleucine (Table 1), starting at a higher level than suggested above. Values for ideal digestible isoleucine for layers ranged from 79 to 86 [8].

As branched-chained amino acids (BCAAs), isoleucine and valine are important for maintaining gut immunity, antioxidant capacity, and critical metabolic processes [7,9,10]. Isoleucine is essential for growth, optimum egg mass, and egg production [11,12]. BCAAs are thought to be important in egg production due to regulation of fatty acid metabolism in the liver where production of lipoprotein could be rate-limiting for egg yolk formation [13]. It is metabolized extra-hepatically in skeletal muscles by BCAA transaminase, competes for transfer across the blood-brain barrier, and is transferred through the cellular membrane via the same pathway as other BCCAs [14,15].

Investigating appropriate quantities of available amino acids such as isoleucine will likely continue due to the world-wide increase in poultry and egg production [16,17]. Shivazad et al. suggested determination of the appropriate quantity of available synthetic amino acids due to less output of nitrogen and positive outcomes for environmental health [11]. Bregendahl et al. suggested ideal amino acid ratios relative to lysine irrespective of source for young White Leghorn layers [6]. Liu and Selle noted that the digestive dynamics of available synthetic amino acids differ from that of protein-bound amino acids [18]. All of these observations should be considered as investigators include new low cost sources of protein with varying quantities of BCAA's in diets.

During feed formulation, the quantity of amino acids in less expensive protein sources can be determined and supplemental synthetic amino acids such as isoleucine can be used to meet the requirement; however, as with protein-bound amino acids, the appropriate requirement for synthetic amino acids is not clear. Leucine requirements are met by various protein sources in the diet while valine and isoleucine are often most examined [19]. For BCAAs, requirements are important because antagonism can occur, depressing performance [20,21].

The availability of synthetic isoleucine provides an opportunity to investigate its requirement in diets containing inexpensive sources of protein bound amino acids [17,20–22]. In the present work, we explored a range of levels (0.66% to 0.84%) for total isoleucine (bound plus synthetic) in the diet to understand how various performance and egg quality parameters for 23- to 30-week-old Lohmann LSL LITE layers were affected generally and to ascertain a specific level of isoleucine for further study with other BCAA's.

## Results and discussion

Levels (% L- Isoluecine) represents the total quantity of the amino acid in each diet.

### Performance (Table 2)

Performance values (feed intake, egg production, mass, and layer weight) for the Control (0.66% isoleucine) were similar to expected values for Lohmann LSL-Lite layers [23].

**Table 1. Digestible amino acid intake for isoleucine from five layer producers.**

| Hy-line 36 (Hy-line International, 2020)[1] | Hy(Hy-line brown Hy-line International, 2020)[2] | Lohmann brown lite (Tierzucht 2017)[2] | Lohmann brown classic (Tierzucht 2017)[2] | Isa brown (Isa Brown, 2011)[3,4] |
|---|---|---|---|---|
| 640[5] | 656[5] | 540[5] | 570[5] | 767[5] |

[1]First egg to production drops 2% below peak egg production (approximately 18–37 wk).

[2]50% egg production to maximum egg production (approximately 21–40 wk).

[3]From 2% eggs production to 28 wk of age.

[4]Based on a feed intake of 105 g/bird/day.

[5]mg/bird/day.

**Table 2. Isoleucine level and performance$^@$ of layers for 23–30 weeks.**

| Isoleucine (%) | Lys:Ile$^{\#}$ | Egg Production (%) | Egg Mass$^{\#\#}$ (g) | Feed Intake (grams/day) | FCR$^{\#\#\#}$ | Weight Gain$^{\#\#\#\#}$ (g) |
|---|---|---|---|---|---|---|
| 0.66* | 79.5 | 91.66[ab] | 50.88[bc] | 111.57[ab] | 1.51[a] | 16.33 |
| 0.69 | 83.1 | 92.17[ab] | 51.35[ab] | 111.90[a] | 1.51[a] | 13.29 |
| 0.72 | 86.7 | 92.37[a] | 52.068[a] | 110.67[bc] | 1.45[b] | 9.45 |
| 0.75 | 90.4 | 90.30[c] | 51.18[b] | 108.47[d] | 1.52[a] | 11.16 |
| 0.78 | 93.9 | 91.05[bc] | 50.19[c] | 109.63[c] | 1.49[ab] | 13.18 |
| 0.81 | 97.6 | 91.66[ab] | 50.84[bc] | 111.59[ab] | 1.50[a] | 13.35 |
| 0.84 | 98.8 | 91.08[bc] | 51.38[ab] | 110.14[c] | 1.44[b] | 20.14 |
| SE | | 2.34 | 1.45 | 2.06 | 0.08 | 28.43 |
| Week | | | | | | |
| 23 | | 79.96[c] | 43.51[g] | 106.81[c] | 1.68[a] | 9.79[bc] |
| 24 | | 87.07[b] | 47.25[f] | 106.87[c] | 1.54[b] | 21.98[b] |
| 25 | | 93.66[a] | 50.13[e] | 107.51[c] | 1.48[c] | 3.83[cd] |
| 26 | | 93.94[a] | 51.43[d] | 110.45[b] | 1.49[c] | 51.73[a] |
| 27 | | 93.79[a] | 52.87[c] | 115.43[a] | 1.41[d] | 3.54[d] |
| 28 | | 94.52[a] | 54.23[b] | 116.48[a] | 1.49[bc] | 3.95[cd] |
| 29 | | 94.77[a] | 55.08[a] | 110.14[b] | 1.39[d] | 7.48[bcd] |
| 30 | | 94.03[a] | 54.51[ab] | 110.85[b] | 1.42[d] | 5.50[bcd] |
| SE | | 1.77 | 1.095 | 1.56 | 0.06 | 21.49 |
| Level | | 0.02 | 0.0002 | <0.01 | < 0.01 | 0.863 |
| Week | | < 0.01 | < 0.01 | < 0.01 | <0.01 | < 0.01 |
| L*W** | | < 0.01 | < 0.01 | < 0.01 | 0.64 | 0.4627 |
| Linear | | 0.07 | 0.0311 | 0.004 | 0.04 | 0.5911 |
| Quadratic | | 0.58 | 0.0818 | 0.001 | 0.41 | 0.1573 |
| Cubic | | 0.25 | 0.4022 | 0.94 | 0.014 | 0.999 |

$^@$Mean (7 diets x 7 replicates x 10 birds per replicate) with different superscripts within a column differ significantly (P ≤ 0.05).

$^{\#}$Lysine: Isoleucine.

$^{\#\#}$ Hen week production (%) multiplied by egg weight (g) and divided by 100.

FCR, kg feed consumed per 12 eggs produced.

$^{\#\#\#\#}$Weight gain over period of study.

*Control.

**Interaction of level × week.

Our results showed that 0.75% isoleucine produced less eggs when compared to 0.66–0.72%, and 0.81% isoleucine, but similar quantities compared to 0.78% and 0.84%. Thus, level decreased production with a linear trend. Dong et al. observed no change in egg production [or any performance measurement (egg mass, feed intake, FCR)] when feeding 0.54% to 0.94% digestible isoleucine to 28-week-old Lohmann Brown Layers in 0.10 intervals for an additional 12 weeks (with a one week acclimation period) [7]. Da Rocha et al. reported a quadratic effect on production when feeding isoleucine:lysine ratios (0.73:1, 0.78:1, 0.83:1, 0.88:1, 0.93:1 and 0.98:1) to Hy-line W-36 layers (24- to 40-week-old) and suggested a isoleucine:lysine ratio of 0.84:1 [4].

With increasing age (week), egg production significantly increased; level by week (L*W) was also significant. As noted by Bell and Weaver (2002), as layers age (week), production decreased [24].

Egg mass was affected by level with linear and quadratic effects. The statistically greatest egg mass (52.068 ± 1.45 g) occurred at 0.72% isoleucine and produced 2.32% more mass than the

Control (50.88 ± 1.45 g). Peganova and Elder reported that greater than 1.0% isoleucine reduced daily egg mass for 24- to 32-week-old and 46- to 54-week-old Lohmann Brown layers [20]. When feeding Hy-Line W-36 (26- to 34-week-old), Bregendahl et al. noted that the ideal amino acid ratio for maximum egg mass was lysine (100):isoleucine [6]). Da Rocha reported a gradratic effect of level on egg mass [4].

Feed intake by level was significant with linear (downward) and quadratic responses.

The significantly lowest feed intake was associated with 0.75% isoleucine. At concentrations of greater than 1.05%, isoleucine reduced egg production, egg mass, egg weight, and feed consumption, likely associated with decreased dopamine, noradrenaline, and serotonin [25]. Peganova and Eder also observed that greater than 1.05% isoleucine was associated with a reduction in feed intake for Lohmann Brown laying hens (24- to 32-week-old) [20]. It is not clear if serotonin, as an appetite suppressant, affected feeding behavior at 0.75% isoleucine in our work because greater quantities of isoleucine did not cause the effect. Da Rocha et al. noted a quadratic effect on feed intake by level of isoleucine [4]. While level affected feed intake, there was no effect on bird weight gain, exhibiting temporal fluctuation. Perganova and Eder noted that dietary isoleucine greater than 0.8% caused a reduction in body weight [20].

In the poultry industry, FCR is most often selected as a useful performance variable in determining the most appropriate diet [26,27]. In our work, FCR linearly decreased significantly with level; the response was also cubic. Isoleucine at 0.72% and 0.84% significantly reduced FCR compared to all other levels except 0.78%. Da Rocha et al. reported a quadratic effect of isoleucine level on FCR [4]. Clark et al. compared feeding behavior of selected ISA Brown layers at 25 to 30 weeks of age [27]. Hens with FCR < 1.8 ± 0.02 (high feed efficiency) consumed less feed (as in our work) and preferred a diet with greater ash content and lower gross energy when compared to hens having high FCR > 2.1 ± 0.02 (low feed efficiency). Week was linear for FCR while $L^*W$ was not.

## External egg quality (Table 3)

Level, significant for egg weight, had a quadratic effect; the greatest egg weight was produced by 0.69%, 0.81%, and 0.84% isoleucine with shared significance at 0.78%. Profit may be made if producers can consistently obtain appropriate amino acid ratios and a lysine:isoleucine at 1:0.69% with low cost feed ingredients while producing eggs with greater weight for markets where this parameter is the most important factor. Da Rocha et al. observed that increasing isoleucine:lysine (0.59, 0.63, 0.67, 0.71, 0.75, and 0.79) did not affect egg weight for Hy-line W36 laying hens (24- to 40-week-old) [4].

A linear downward trend was produced by level for egg shape. Eggs with a shape index < 76 may not withstand processing, packaging, and transportation due to reduced shell strength [28]. Duman et al. observed no effect of shape index on breaking strength [29]. While shape index may be important for commercial producers, it is likely less important in production for local markets or backyard production.

For specific gravity, level caused a linear (upward) effect, greatest at 0.84% (1.0888 ± 0.0043) compared to the Control (1.0820 ± 0.0043); there were quadratic and cubic effects. As specific gravity increases, shell weight should increase [30]. However, proportional shell weight was not significantly affected by level in our work.

Our results indicated that shell thickness was significant for level with quadratic and cubic effects. Isoleucine at 0.72% produced the lowest egg shell thickness (0.379 mm + 0.015 SE) that was between the medium (0.33–0.36 mm) and thick (0.39–0.41 mm) category as reported by Ketta and Tůmová [31]. More research is needed to elucidate how isoleucine level affects eggshell thickness. It is known that after fermentation of eggshells to

**Table 3. Isoleucine level and external egg quality@ of layers for 23–30 weeks.**

| Isoleucine (%) | Lys:Ile# | Egg Weight (g) | Egg Shape Index | Specific Gravity | Shell Thickness (mm) | Proportional Egg Shell Weight## (%) |
|---|---|---|---|---|---|---|
| 0.66* | 79.5 | 55.96[b] | 74.69 | 1.0820[bc] | 0.399[a] | 10.386 |
| 0.69 | 83.1 | 56.44[a] | 75.17 | 1.0817[c] | 0.398[ab] | 10.537 |
| 0.72 | 86.7 | 55.35[d] | 75.18 | 1.0838[bc] | 0.379[c] | 10.452 |
| 0.75 | 90.4 | 55.47[cd] | 74.61 | 1.0838[bc] | 0.391[b] | 10.540 |
| 0.78 | 93.9 | 55.77[bc] | 74.75 | 1.0838[bc] | 0.400[a] | 10.304 |
| 0.81 | 97.6 | 55.99[b] | 74.70 | 1.0841[b] | 0.393[ab] | 10.466 |
| 0.84 | 98.8 | 55.82[b] | 74.95 | 1.0888[a] | 0.398[ab] | 10.530 |
| SE | | 0.57 | 1.034 | 0.0043 | 0.015 | 0.4780 |
| Week | | | | | | |
| 23 | | 54.42[de] | 75.35[ab] | 1.0669[e] | 0.387[d] | 10.949[ab] |
| 24 | | 54.27[e] | 75.52[a] | 1.0686[e] | 0.424[a] | 10.257[cd] |
| 25 | | 53.52[f] | 74.99[abc] | 1.0886[bc] | 0.388[d] | 9.8538[e] |
| 26 | | 54.73[d] | 74.55[cd] | 1.0965[a] | 0.411[b] | 11.156[a] |
| 27 | | 56.25[c] | 74.82b[cd] | 1.0950[a] | 0.400[c] | 10.45[c] |
| 28 | | 57.37[b] | 74.55[cd] | 1.0898[b] | 0.417[ab] | 10.87[b] |
| 29 | | 58.12[a] | 74.37[d] | 1.0874[c] | 0.401[c] | 10.12[d] |
| 30 | | 57.97[a] | 74.25[d] | 1.0788[d] | 0.326[e] | 10.019[de] |
| SE | | 0.43 | 0.78 | 0.0032 | 0.011 | 0.361 |
| Level | | <0.01 | 0.379 | <0.01 | <0.01 | 0.436 |
| Week | | <0.01 | <0.01 | <0.01 | <0.01 | <0.01 |
| L*W ** | | <0.01 | 0.0892 | <0.01 | <0.01 | <0.01 |
| Linear | | 0.12 | 0.059 | <0.01 | 0.6235 | 0.7596 |
| Quadratic | | 0.002 | 0.15 | 0.0277 | 0.0006 | 0.861 |
| Cubic | | 0.72 | 0.44 | 0.0302 | 0.0246 | 0.1000 |

@Means (7 diets x 7 replicates x 3 eggs per replicate) with different superscripts within a column differ significantly (P ≤ 0.05).

#Lysine: Isoleucine.

##Dried shell weight as percent of total egg weight.

*Control.

**Interaction of level × week.

produce alkaline protease, the protein hydrolyzate contained several essential amino acids including arginine, isoleucine, leucine, lysine, methionine, phenylalanine, and valine along with smaller quantities of cysteine; thus, eggshells have been proposed as a viable protein source [32]. As well, changes in epimerization of isoleucine in eggshell have been used to date the range of age for archeological sites [33]. All other levels of isoleucine in the present work where layers were raised in a cage system with no enrichment produced eggshells in the thick range as classified by Ketta and Tůmová for their eggs in a caged system with enrichment or pens with litter [31].

In their investigation of worldwide eggshell uniformity, Sun et al. noted that eggs were thickest (0.367 + 0.023mm) at the sharp end and thinnest at the blunt end (0.341 + 0.025 mm) [34]. Mean thickness of the shells across seven locations with and without the membrane was 0.369 ± 0.021 and 0.356 ± 0.022 mm, respectively [34]. Thus, all of the shells in our study met or exceeded mean thickness. All external quality measurements were significantly affected by week. L*W also affected all measurements excluding shape index where there was a linear trend.

### Internal egg quality (Table 4)

Level had no effect on Haugh unit whereas week and L*W were significant due to fluctuations. Proportional yolk weight had linear, quadratic, and cubic effects due to level; the highest level of isoleucine (0.84%) produced the greatest value (26.26 ± 0.91g) compared to all other levels; and the yolk:albumen ratio (41.53 ± 0.0205g, with significant liner, quadratic, and cubic effects) was also greatest at this quantity of isoleucine. Perhaps 0.84% isoleucine increased deposition of lipid in yolk [13].

Level affected proportional albumen weight, with fluctuations, causing significant quadratic and cubic responses. Proportional albumen results differed from those of Da Rocha et al. [4] who found insignificance for isoleucine levels (0.59 to 0.79%) when feeding Hy-Line W36 layers (24- to 40-week-old). Yolk index was significant for level with linear and cubic effects.

In the present work, yolk color was not affected by level. As we observed, Da Rocha et al. reported insignificance for isoleucine levels (0.59 to 0.79%) on yolk color [4]. Although fat soluble carotenoids may not have been the same for our diets and those of Da Rocha et al.,

**Table 4. Isoleucine level and internal egg quality[@] of layers for 23 to 30 weeks.**

| Isoleucine (%) | Lys: Ile[#] | Haugh Unit | Proportional Yolk Weight (%)[##] | Proportional Albumen Weight (%)[##] | Yolk Index | Albumen Index | Y:A[###] | Yolk Color |
|---|---|---|---|---|---|---|---|---|
| 0.66[*] | 79.5 | 94.26 | 25.36[bc] | 64.258[ab] | 43.246[ab] | 12.09 | 39.65[bc] | 3.464 |
| 0.69 | 83.1 | 94.12 | 25.24[bc] | 64.221[abc] | 42.834[bcd] | 12.26 | 39.5[bc] | 3.381 |
| 0.72 | 86.7 | 94.53 | 25.54[b] | 64.035[bc] | 43.061[abc] | 12.05 | 40.18[b] | 3.351 |
| 0.75 | 90.4 | 93.74 | 25.17[bc] | 64.256[ab] | 42.450[de] | 12.23 | 39.21[bc] | 3.357 |
| 0.78 | 93.9 | 94.15 | 24.92[c] | 64.782[a] | 42.622[cd] | 12.40 | 38.71[c] | 3.434 |
| 0.81 | 97.6 | 94.65 | 25.59[b] | 63.950[bc] | 43.341[a] | 12.26 | 40.16[b] | 3.304 |
| 0.84 | 98.8 | 94.09 | 26.26[a] | 63.662[c] | 41.954[e] | 12.06 | 41.53[a] | 3.488 |
| SE | | 2.12 | 0.91 | 1.11 | 0.95 | 0.67 | 0.0205 | 0.37 |
| Week | | | | | | | | |
| 23 | | 92.07[e] | 25.69[ab] | 63.89[bc] | 43.232[b] | 11.59[e] | 40.53[ab] | 2.939[c] |
| 24 | | 87.46[f] | 25.25[bc] | 64.491[ab] | 40.703[d] | 10.19[f] | 39.39[bcd] | 3.048[c] |
| 25 | | 92.09[e] | 25.07[c] | 65.073[a] | 42.087[c] | 11.58[e] | 38.71[d] | 3.578[b] |
| 26 | | 94.97[cd] | 25.09[c] | 63.745[c] | 43.321[b] | 12.51[c] | 39.36[dc] | 3.769[ab] |
| 27 | | 97.07[b] | 24.94[c] | 64.607[a] | 44.795[a] | 12.95[b] | 38.77[d] | 3.728[b] |
| 28 | | 99.78[a] | 25.38[bc] | 63.758[c] | 43.184[b] | 14.09[a] | 40.08[abc] | 3.973[a] |
| 29 | | 94.33[d] | 25.95[a] | 63.919[bc] | 42.506[c] | 12.02[d] | 40.77[a] | 3.143[c] |
| 30 | | 95.99[bc] | 26.13[a] | 63.847[c] | 42.469[c] | 12.59[bc] | 41.17[a] | 3.0[c] |
| SE | | 1.6 | 0.69 | 0.842 | 0.718 | 0.507 | 0.0155 | 0.280 |
| Level | | 0.764 | <0.01 | 0.013 | <0.01 | 0.397 | <0.01 | 0.483 |
| Week | | <0.01 | <0.01 | <0.01 | <0.01 | <0.01 | <0.01 | <0.01 |
| L*W[**] | | <0.01 | 0.4070 | 0.017 | <0.01 | <0.01 | 0.2456 | 0.0002 |
| Linear | | 0.934 | 0.0024 | 0.17 | 0.0005 | 0.6936 | 0.0076 | 1.0 |
| Quadratic | | 0.834 | <0.01 | 0.04 | 0.6048 | 0.1931 | 0.0005 | 0.1285 |
| Cubic | | 0.740 | 0.0056 | 0.02 | 0.0020 | 0.2247 | 0.0047 | 0.9171 |

[@]Means (7 diets x 7 replicates x 3 eggs per replicate) with different superscripts within a column differ significantly (P ≤ 0.05). % isoleucine as formulated.

[#]Lys/Ile for lysine: Isoleucine.

[##] Egg albumen or yolk weight/egg weight × 100.

[###]Yolk: Albumen.

[*]Control.

[**]Interaction of level × week.

insignificance of level on yolk color in both studies likely indicated constancy of carotenoids across the diets of each individual study [4]. Dong et al. found no effect of feeding 0.54% to 0.94% isoleucine on internal quality measurements (albumen height, Haugh units, and yolk color) [7].

Week was significant for all internal measurements. L*W affected proportional albumen weight, yolk index, albumen index, and yolk color.

**Comparative performance and external/internal egg quality measurements.** The effect of level on performance parameters and external/internal egg quality measurements reported in the present study did not always agree with findings of other investigators even when the same range of isoleucine was added to layer diets. Age of layers, length of feeding, genetics, environmental factors, or imbalances of amino acid may be causative factors. As well, as noted above, Liu and Selle reported that the digestive dynamics of available synthetic amino acids differ from that of protein-bound ones [35]. More work is needed to unravel the digestive dynamics because protein bound amino acids (from inexpensive sources) and available synthetic ones likely will be fed to layers more often as in the present work. Other investigators suggested that due to possible imbalances, all BCAA's should be monitored when investigating any one of them in commercial avian diets [36]. Moreover, as we reported, L*W often affects many parameters. Perhaps large scale coordinated investigations across several laboratories with several types of layers over their complete laying cycle are needed to find the most appropriate level of isoleucine, valine, and leucine for layers.

## Biochemical analyses and crude protein digestibility coefficient (Table 5)

A downward linear trend for serum protein was observed for level. This result did not agree with those of Tewe who reported correlation of both protein quantity and quality with total serum protein [37].

For globulin, there was a significant downward linear trend. Levels of isoleucine from 0.66% (0.81 ± 0.16 g/dL) to 0.75% (0.85 ± 0.16 g/dL) produced a statistically similar response for globulin that was different from that for 0.84% (0.35 ± 0.16 g/dL). The reduction in globulins at 0.84% isoleucine was 58% compared to the Control (0.81 g/dL + 0.16 SE). Harlap et al. studied blood proteins in laying hens (Lohmann Whites, 26- to 80-week-old) [38]. They reported that the quantity of globulins was not related to the productive period but rather depended on a reduction in defenses with the greatest decrease at 80 weeks of age. At 30 weeks old, our control layers had globulin levels at or above 0.81g/dL (0.62% isoleucine to 0.75% isoleucine) that were gradually reduced to 0.35 g/dL at 0.84% isoleucine. The reduction of globulins indicated that increasing the isoleucine level beyond 0.75 may be related to a reduction of immunological defenses of the layers. This finding may be important; however, it is not clear if findings for Lohmann Whites are similar to all other breeds of layers, including LSL-LITE layers. Clearly, more research on these findings is indicated.

For glucose, level produced a downward linear response with quadratic and cubic trends. For ileal CP digestibility, there was a significant cubic effect for level. At 0.81% isoleucine, the coefficient was 19.65% less than that for the Control (82.71% ± 6.26 SE).

Total serum albumin, digestibility, and HI titers were not affected by level. Contrary to our work, Dong et al. reported a quadratic effect for isoleucine levels on serum albumin [7]. The genetic variation or subtype, susceptibility of host, age, breed, and environment (such as environmentally controlled houses versus open houses) are factors likely affecting ND titers of layers in our work and that of others [39]. Additionally, no affect for ND titers associated with level of isoleucine in our work could be explained by presence of specific circulating antibodies produced during low levels of infections associated with low endemic levels of NDV in the

**Table 5. Isoleucine level, serum biochemistry, and ileal digestibility of crude protein values[@] in layers for 23 to 30 weeks.**

| Isoleucine (%) | Lys: Ile[#] | Total Serum Protein | Serum Albumin | Globulin | Glucose | CP Digestibility Coefficient[##] | HI Titer | |
|---|---|---|---|---|---|---|---|---|
| | | —————————————————g / dL———————— | | | mg / dL | (%) | ND | H9 |
| 0.66 [*] | 79.5 | 3.68 | 2.87 | 0.81[a] | 170.14[ab] | 82.713[a] | 8.57 | 8.71 |
| 0.69 | 83.1 | 3.65 | 3.08 | 0.84[a] | 208.29[a] | 77.743[ab] | 8.28 | 8.42 |
| 0.72 | 86.7 | 3.30 | 2.85 | 0.82[a] | 220.71[a] | 79.983[ab] | 7.85 | 8.14 |
| 0.75 | 90.4 | 3.35 | 3.00 | 0.85[a] | 194.71[a] | 79.143[ab] | 7.42 | 9.00 |
| 0.78 | 93.9 | 3.20 | 2.72 | 0.70[ab] | 111.43[bc] | 68.700[ab] | 8.00 | 8.42 |
| 0.81 | 97.6 | 3.11 | 2.81 | 0.54[ab] | 112.86[bc] | 66.457[b] | 8.57 | 8.57 |
| 0.84 | 98.8 | 3.00 | 2.81 | 0.35[b] | 104.43[c] | 77.563[ab] | 8.71 | 7.71 |
| SE | | 0.25 | 0.23 | 0.16 | 20.61 | 6.26 | 0.80 | 0.64 |
| *P*-value | | | | | | | | |
| Level | | 0.07 | 0.77 | 0.03 | < 0.01 | 0.1554 | 0.67 | 0.55 |
| Linear | | 0.01 | 0.39 | 0.09 | < 0.01 | 0.1266 | 0.74 | 0.85 |
| Quadratic | | 0.22 | 0.79 | 0.73 | 0.06 | 0.9585 | 0.34 | 0.32 |
| Cubic | | 0.69 | 0.69 | 0.57 | 0.07 | 0.0456 | 0.61 | 0.37 |

[@]Means (7 diets x 7 replicate x 2 birds per replicate) with different superscripts within a column differ significantly (P ≤ 0.05).

[#]Lysine: Isoleucine.

[##]CP Digestibility Coefficient as ileal digestibility for crude protein.

[*]Control.

poultry industry worldwide [40]. As well, H9N2 virus was detected in 2015 in a Pakistani poultry worker, suggesting that H9 could be transmitted by staff to layers [41]. To date, NDV and AI are prevalent in Pakistan and are included in the commercial vaccination schedule [42].

## Conclusion

For the purpose of our work, it was important to determine a specific level of isoleucine as an important parameter to compare to other BCAA's. Isoleucine levels at 0.72% and 0.84% produced low FCR of 1.45 and 1.44, respectively, but shared significance with 0.78% (1.49). Also, at 0.72% isoleucine, other egg production, egg quality, and biochemical measurements were in the medium to high range. Considering cost of synthetic isoleucine as a dietary ingredient along with the low FCR, 0.72% was chosen as the constant for future studies for comparison of valine and/or leucine levels in layer diets.

## Materials and methods

LSL-LITE layers (n = 490, at 23 to 30 weeks) were allotted to 7 treatments × 7 replicates × 10 layers per replicate. Two adjoining cages (60 x 63.5 cm, five birds each) constituted a replicate. Hens were fed the Control (basal diet + 0.66% L-isoleucine, as calculated, Table 6) and diets containing calculated total percentages of L-isoleucine at 0.69, 0.72, 0.75, 0.78, 0.81, and 0.84%.

Feed in mash form and fresh water were provided *ad libitum*. Hens were housed in a semi-controlled environment with a constant 16L:8D photoperiod (10–15 Lux) where day to night temperatures were a maximum of 28.71 ± 0.12°C and minimum of 26.05 ± 0.11°C during August and September. The study was conducted on a commercial layer farm (Lahore,

**Table 6. Nutrient composition[1] of control diet for layers at 23 to 30 weeks.**

| Ingredients | Control feed | Nutrients % | Calculated[1] | Analyzed[2] |
|---|---|---|---|---|
| Corn | 54.00 | Dry Matter | 90.32 | 90.27 |
| Rice Tips | 6.00 | ME (Kcal / Kg) | 2728.00 | -- |
| Soybean Meal | 12.00 | ME (MJ / Kg) | 11.41 | |
| Canola Meal | 5.00 | Crude Protein | 16.77 | 17.28 |
| Sunflower Seed Meal | 5.00 | Ether Extract | 3.06 | -- |
| Corn Gluten Meal 60% | 2.00 | Ash | 2.27 | -- |
| Guar Meal | 2.00 | Crude Fiber | 4.33 | -- |
| Poultry By-product Meal | 2.00 | Calcium | 3.56 | -- |
| Canola Oil | 0.70 | Dig phosphorus | 0.42 | -- |
| CaCO$_3$ | 8.00 | Total phosphorus | 0.67 | -- |
| Di-Calcium Phosphate | 1.80 | Sodium | 0.17 | -- |
| L-Lysine SO$_4$ | 0.40 | Potassium | 0.62 | -- |
| DL-Methionine | 0.15 | Chloride | 0.16 | -- |
| L-Threonine | 0.10 | Lysine | 0.83 | 0.91 |
| L-Tryptophan | 0.05 | Methionine | 0.42 | 0.39 |
| **L-Isoleucine[3]** | **0.10** | Threonine | 0.62 | 0.70 |
| NaHCO$_3$ | 0.40 | Tryptophan | 0.18 | 0.19 |
| Vitamins Minerals Premix[4] | 0.30 | Cysteine | 0.26 | 0.31 |
| | | Met + Cys | 0.67 | 0.71 |
| | | Arginine | 0.95 | 1.08 |
| | | Valine | 0.67 | 0.88 |
| | | **Isoleucine** | **0.66** | **0.78** |
| | | Leucine | 1.29 | 1.38 |
| | | Histidine | 0.37 | 0.43 |
| | | Phenylalanine | 0.69 | 0.74 |
| | | Linoleic Acid | 1.66 | -- |
| | | Na + K--Cl (mEq / Kg) | 214.31 | -- |

[1]Calculated column estimated as digestible during feed formulation. For diets, quantities of synthetic isoluecine were added to yield total L-isoluecine at 0.69, 0.72, 0.75, 0.78, 0.81, and 0.84%.

[2]Analyzed for total amino acids (Amino Lab® Evonik SEA Pte. Ltd. Singapore—Lab code: SG16-0000618-001) [17].

[3]Purity > 98.0%.

[4] Met or exceeded the minimum NRC requirements [43].

Pakistan) with uniform production performance (72.14% ± 1.66) and body weight (1.4 Kg ± 0.006).

The experimental protocol (including handling, care, and humane end point by cervical dislocation for ileal digestibility) was approved by the Ethical Review Committee, University of Veterinary and Animal Sciences, Lahore, Pakistan. As required in the approved protocol, birds were monitored twice daily for signs of illness (lethargy, not frequently consuming food, injury, or distress). There was no mortality. Hens remaining at the end of the study were returned to the layer farm.

Daily egg production, egg weight, feed intake, body weight gain, and FCR were determined. External measurements included egg weight, egg mass, specific gravity, shape index (egg width/egg length × 100), egg shell weight, and egg shell thickness [44–46]. Internal measurements included egg yolk color (Roche Yolk Color Fan) egg yolk index (egg yolk height/egg

yolk diameter × 100), albumen index (albumen height/average egg albumen width × 100), and yolk to albumen ratio (egg yolk weight/weight of albumen) [47–49]. Proportional albumen ratio (egg albumen weight/egg weight × 100) was also determined. Haugh unit was calculated as:

HU = 100 × log (albumen height—1.7+ egg weight ^ 0.37 +7.57 [49].

Celite (2% of diet) was added three days before the end of the study. On the last day of the study, blood (~ 4 ml) was taken from the jugular vein of two birds per replicate. Blood samples were collected in gel vacutainers (gel clot activators-ImuMed$^{®}$) [50]. Serum concentrations of total protein and glucose were measured spectrophotometrically (Merck, USA, Human Kits, Eggenstein, Germany). Newcastle Disease Virus (NDV) and H9 (subtype of avian influenza) antibody titers were determined by procedures of Rubbani et al. [51].

After blood samples were obtained, cervical dislocation was performed to obtain ileal digesta and analyzed [7,52–58]. and analyzed. Total nitrogen in feed and digesta (oven dried, 55–60˚C) were used to calculate ileal digestibility of protein by Kluth and Rodehutscord as:

Digestibility Coefficient (%) = 100–100 x [(Celite in Diet x Protein in Digesta)/Celite in Digesta x Protein in Diet] [59].

## Statistical analysis

Data were analyzed by a one way ANOVA under a Completely Randomized Design and evaluated using orthogonal polynomials for linear, quadratic, and cubic responses [60]. Feed intake, FCR, body weight change, and egg quality parameters were analyzed by repeated measures (SAS, version 9.1, 2021) using PROC GLM [61]. Significance at P ≤ 0.05 was determined; P ≤ 0.10 was noted [62].

## Acknowledgments

Authors acknowledge EVONIK (Animal Nutrition, Singapore) for support of the study and analysis of amino acids.

## Author Contributions

**Conceptualization:** S. Ullah, Y. A. Ditta, A. J. King, T. N. Pasha, A Mahmud, K. A. Majeed.

**Data curation:** S. Ullah, A Mahmud.

**Funding acquisition:** Y. A. Ditta.

**Methodology:** T. N. Pasha, K. A. Majeed.

**Supervision:** A. J. King.

**Writing – original draft:** Y. A. Ditta.

**Writing – review & editing:** A. J. King.

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
