## [Decision Letter · Decision Letter 0]

15 Jan 2021

PONE-D-20-35024

Varying digestible isoleucine level to determine effects on performance, egg quality, serum biochemistry, and ileal protein digestibility in diets of young laying hens

PLOS ONE

Dear Dr. King,

Thank you for submitting your manuscript to PLOS ONE. After careful consideration, we feel that it has merit but does not fully meet PLOS ONE’s publication criteria as it currently stands. Therefore, we invite you to submit a revised version of the manuscript that addresses the points raised during the review process.

We look forward to receiving your revised manuscript.

Kind regards,

Mahmoud A.O. Dawood, PhD

Academic Editor

PLOS ONE

Journal Requirements:

2. In your Methods section, please provide additional details regarding the animals used in your study and ensure you have described the source.

For more information regarding PLOS' policy on materials sharing and reporting, see https://journals.plos.org/plosone/s/materials-and-software-sharing#loc-sharing-materials

3. In your Methods section, please state the volume of the blood samples collected for use in your study.

'Research was partially supported by EVONIK Animal Nutrition, Singapore. Evonik analyzed amino acids in each diet.'

'The funders had no role in study design, data, collection, decision to publish, or preparation of the manuscript.'

b. Additionally, because some of your funding information pertains to commercial funding, we ask you to provide an updated Competing Interests statement, declaring all sources of commercial funding.

In your Competing Interests statement, please confirm that your commercial funding does not alter your adherence to PLOS ONE Editorial policies and criteria by including the following statement: "This does not alter our adherence to PLOS ONE policies on sharing data and materials.” as detailed online in our guide for authors  http://journals.plos.org/plosone/s/competing-interests. 

If this statement is not true and your adherence to PLOS policies on sharing data and materials is altered, please explain how.

c. Please include the updated Competing Interests Statement and Funding Statement in your cover letter. We will change the online submission form on your behalf.

6. Please include captions for your Supporting Information files at the end of your manuscript, and update any in-text citations to match accordingly. Please see our Supporting Information guidelines for more information: http://journals.plos.org/plosone/s/supporting-information

Reviewers' comments:

Reviewer's Responses to Questions

**Comments to the Author**

1. Is the manuscript technically sound, and do the data support the conclusions?

Reviewer #1: Partly

Reviewer #2: Yes

2. Has the statistical analysis been performed appropriately and rigorously? 

Reviewer #1: No

Reviewer #2: Yes

3. Have the authors made all data underlying the findings in their manuscript fully available?

Reviewer #1: Yes

Reviewer #2: Yes

4. Is the manuscript presented in an intelligible fashion and written in standard English?

Reviewer #1: No

Reviewer #2: Yes

5. Review Comments to the Author

Reviewer #1: Reviewer’s comments on Manuscript no PONE-D-20-35024.

The topic is very interesting and does allow knowledge transfer in the field of avian nutrition. However, the manuscript lacks certain details, which must be added. Major revision is required before it can be accepted for publication. The comments below will allow the authors to improve their manuscript:

Abstract:

Line 1 & 2: In control diet, 0.10 isoleucine was added to achieve 0.66% isoleucine levels. Mention the supplemental levels of isoleucine for all diets to achieve 0.66 to 0.84% isoleucine in complete feed.

Line 3 & 4: The authors must mention the control diet's crude protein value along with the ME values.

Line 6 &7: What is meant by weight change? Do you mean weight gain? Rephrase the term to clarify.

Line 12-14: Authors need to do a regression analysis to confirm the optimum levels of isoleucine for commercially important parameters. There may be different optimum levels for different parameters.

Introduction:

The introduction is not clear and needs to be elaborated and rewritten to justify higher isoleucine levels? What has changed, which demands higher levels?

Line 7-8: “Researchers have proposed optimum isoleucine levels to be from 550 to 660mg/hen /day”: But it is not clear how much of the supplemental isoleucine is needed in the diet to fulfil this requirement? This recommendation is for which breeds and what ages??

Line 8-9: Sentence needs revision. What do authors mean by genetic variation and age groups?

Results:

Table 1: was it for 20 to 30 weeks? Or 23 to 30 weeks?

Table 1: Units for Egg production and Feed intake is missing?

Table1: Change in bird weight? Not clear? I think authors means initial at minus final wt. If that is the case, then it should be mentioned as weight gain.

Table 1: Egg mass data is missing? Must add.

Page 6: line 2: Bodyweight change was significant for the week? Did it increase or decrease with age?

Page 6: Line 3: numerical differences mean nothing? The level of egg production was statistically lower for all treatments fed 0.75 to 0.84% supplemental isoleucine? Revise sentence to clarify what authors are trying to highlight?

Line 4: Highest numerical FCR? This is not true. Numerical differences should not be highlighted. Revise the sentence.

Table 2: Line 5-12: Needs revision: The authors mention that levels and weeks were significantly affected for external egg quality but did not explain the major differences, were they good or bad differences? The optimum level of isoleucine was different for each parameter. For example, For level as a factor, significantly higher egg weight was achieved at 0.69%, higher specific gravity was achieved at 0.84% isoleucine, and shell thickness was highest at 0.78% isoleucine.

Page 7: Internal egg quality (Supplement, Table 1):

It is not clear from the table what does propositional yolk and albumen weight means? Propositional to what? Please clarify?

For proportional yolk weigh, the optimum level of isoleucine was 0.84%. For proportional albumen weigh, the optimum level was 0.78%. The optimum level was 0.81% for yolk index, Y: A the optimum level was 0.84%. Can they all be mentioned in the text?

Regression analysis is required to determine the optimum level.

Table3: Were these levels within the expected physiological range? Not clear. Is an increase or decrease of these significant levels a good or a bad indicator? The whole section needs to be rewritten to clarify the differences between levels? Again, which level of isoleucine resulted in desired effects?

Table 3: A word is missing under quadratic (See the last row of the table?

Page 8: Discussion:

Line 11-12: Was Dong referring to egg production? Please mention.

Line 13: greater quantities? Be specific, mention what were the optimum levels for the egg production. Revise sentences.

Line 15-16: Excessive BCAAS? not clear, what is categorised as excessive levels? Revise the sentence. Which level of isoleucine can cause excessive BCAAs??

Line 16-17: Sentence needs revision: The authors need to clarify that Serotonin is an appetite suppressant. Therefore, speculation can be made that reduced feed intake at 0.75% can be associated with isoleucine level. Not clear what happened at 0.81? Please check and amend the sentence.

Page 9:

Line 3 to 4: "when less expensive feeds are used" what are less expensive feeds? Do you mean low CP feeds? Clarify the sentence.

Line 10- "Lowering isoleucine levels" What were the isoleucine levels used in Shim et al? Was the effect due to lowering of CP levels in general or was the effect specific to isoleucine levels? Not clear.

Line 13-14: what were the lower levels of isoleucine used by Dong? Add the information.

Page 10:

Line 2-3: effect of protein sources and levels of isoleucine are two different things. Clarify the sentence.

Line 3-5: Are you comparing isoleucine with Valine? Do you expect both AA's to affect the immunoglobulins in a similar way? or is it a typo? Please check and clarify.

Line 8-10: The CP levels of this study is 16%. It may be that increase in CP level can affect ND titer. There are many other factors that can affect the ND titre? These other factors can be explored to describe your results. The sentence needs clarification.

Line 11: what are "protein deprived chicks"? Not clear.

Line 15-16: Line 14 to 17: needs revision: recommendation for further studies is very vague. Can they be specific?

Material and methods:

• Throughout the text, it should be made clear that isoleucine levels are total and not SID.

• Methods should also specify L- isoleucine quantity added in each diet to meet the required % isoleucine levels.

Page 11:

Line 6: Methods should also specify the type of cage system that was used in this study.

Line 12: the formula used to determine ileal CP digestibility should be provided in the manuscript.

Line 15: Hempe et al: year of publication is missing

Line 16: Victor and Carver, 1936: Reference is too old. Please replace it with any recent reference if possible.

Line 16: Haugh 1938: Reference is missing in the reference list. Again reference too old.

How internal and external egg parameters were analysed? The information must be added in the methods section.

Page 12:

Line 3: Was digesta samples freeze or oven-dried. Add the information.

Statistical analysis:

• For egg production parameters it was factorial design (L*W), how can that be analysed as one way? It has to be a factorial design?

• Data must be analysed as regression analysis to confirm optimum levels of isoleucine for different parameters.

Reviewer #2: This is the review for the manuscript PONE-D-20-35024, entitled “Varying digestible isoleucine level to determine effects on performance, egg quality, serum biochemistry, and ileal protein digestibility in diets of young laying hens” for PLOS ONE Journal.

The present manuscript presents original data, which has been collected using appropriate standard methods. However, Minor concerns still need to be addressed or clarified.

Page 3 (Line 2): Abbreviations should be clarified when first mentioned e.g., AA = Amino acids. Clarify the three limited amino acids (including tryptophan).

Page 9 (Line 12): It is not clear why shell thickness changed with isoleucine supplementation. More explanation is required

Page 10 (Line 13): No discussions for the alteration in serum glucose content.

Page 10 (Line 19): what is 490?

Page 10 (Line 20): "and the Control + 0.69". Control or basal diet?

Page 12 (Lines 2 & 3): Many citations. Use one or two.

6. PLOS authors have the option to publish the peer review history of their article (what does this mean?). If published, this will include your full peer review and any attached files.

Reviewer #1: No

Reviewer #2: No

---

## [Author Response · Author response to Decision Letter 0]

5 Sep 2021

We have added information on data storage and support for research. Information about animals used has been added in Materials and Methods. As well, we have answered all questions/comments of reviewers.

---

## [Decision Letter · Decision Letter 1]

11 Oct 2021

PONE-D-20-35024R1Varying digestible isoleucine level to determine effects on performance, egg quality, serum biochemistry, and ileal protein digestibility in diets of young laying hensPLOS ONE

Dear Dr. King,

Thank you for submitting your manuscript to PLOS ONE. After careful consideration, we feel that it has merit but does not fully meet PLOS ONE’s publication criteria as it currently stands. Therefore, we invite you to submit a revised version of the manuscript that addresses the points raised during the review process.

We look forward to receiving your revised manuscript.

Kind regards,

Mahmoud A.O. Dawood, PhD

Academic Editor

PLOS ONE

Journal Requirements:

Reviewers' comments:

Reviewer's Responses to Questions

**Comments to the Author**

1. If the authors have adequately addressed your comments raised in a previous round of review and you feel that this manuscript is now acceptable for publication, you may indicate that here to bypass the “Comments to the Author” section, enter your conflict of interest statement in the “Confidential to Editor” section, and submit your "Accept" recommendation.

Reviewer #1: All comments have been addressed

Reviewer #2: All comments have been addressed

2. Is the manuscript technically sound, and do the data support the conclusions?

Reviewer #1: Yes

Reviewer #2: Yes

3. Has the statistical analysis been performed appropriately and rigorously? 

Reviewer #1: Yes

Reviewer #2: Yes

4. Have the authors made all data underlying the findings in their manuscript fully available?

Reviewer #1: Yes

Reviewer #2: Yes

5. Is the manuscript presented in an intelligible fashion and written in standard English?

Reviewer #1: Yes

Reviewer #2: Yes

6. Review Comments to the Author

Reviewer #1: The manuscript has been much improved. However, there are some minor corrections which are still required. Please see the comments which needs addressing:

Page 1:

Title: remove the word “digestible” as digestible levels of Isoleucine are not investigated in this study? It’s the L-Isoleucine levels that are studied in this experiment.

Page 2:

Line 16: Further studies to investigate what? Not clear.

Page 3:

Line 10: Table 1: Units are missing?

Page 6:

Line 1: Replace the word “numerically” with “statistically” because the differences were statistically different.

Page 7:

Table 1: wrong numbering: Table on page 3 is also Table 1?

Table 1. Symbol ### is similar for FCR and Weight gain: check and correct. Footnote for how FCR was calculated is missing

Page 9:

Line 12: Fermentation? After which fermentation? Clarify?

Line 17: Reference is missing

Line 19: Incomplete reference. Year is missing in reference?

Page 14:

Line 12: Were differences within an acceptable range? What is a reduction in levels indicating? Good or bad?

Line19: which viral strain?

Page 15:

Rewrite conclusion section:

Line 4 and 5: Delete the sentence. This is not your conclusion.

Line 8: Remove reference from the conclusion section. The authors can use this reference in the discussion section.

Line 11? Further studies to test what? Would you please clarify which “further” studies are required? To investigate what?

Page 15 vs 16: Material and Methods: Line 14 on page 15 says you had 10 layers per replicate, but in Line 13 on Page 16, authors say there is five birds/cage? Please check and confirm if you had 5 or 10 laying hens/replicate?

Page 17:

Line 1: Delete the line “ except for layers …..below.” Culls are not counted as Mortality.

Line 13: what was the level of celite used?

Line 20: Revise sentence. Remove the word “above” and mention that it was ileal digesta.

Reviewer #2: (No Response)

7. PLOS authors have the option to publish the peer review history of their article (what does this mean?). If published, this will include your full peer review and any attached files.

Reviewer #1: No

Reviewer #2: **Yes: **Dr. Mohammed El Basuini

---

## [Author Response · Author response to Decision Letter 1]

10 Nov 2021

Reviewer #1: The manuscript has been much improved. However, there are some minor corrections which are still required. Please see the comments which needs addressing:

Page 1:

Title: remove the word “digestible” as digestible levels of Isoleucine are not investigated in this study? It’s the L-Isoleucine levels that are studied in this experiment.

Digestible has been removed from the title.

Varying isoleucine level to determine effects on performance, egg quality, serum 

biochemistry, and ileal protein digestibility in diets of young laying hens

Page 2:

Line 16: Further studies to investigate what? Not clear.

Considering the low FCR of 1.45 and cost for inclusion as a dietary ingredients, 0.72% isoleucine was chosen for further studies with varying quantities of other branched chain amino acids in diets for young laying hens. 

Page 3:

Line 10: Table 1: Units are missing?

Added 5mg/bird/day.

Page 6:

Line 1: Replace the word “numerically” with “statistically” because the differences were statistically different.

Egg mass was affected by level with linear and quadratic effects. The numerically statistically greatest egg mass (52.068 + 1.45 g) occurred at 0.72% isoleucine and produced 2.32% more mass than the Control (50.88 + 1.45 g).

Page 7:

Table 1: wrong numbering: Table on page 3 is also Table 1?

Table 2. Isoleucine level and performance@ of layers for 23-30 weeks. 

This table number was changed and then all other numbers for tables were changed in the entire manuscript.

Table 1. Symbol ### is similar for FCR and Weight gain: check and correct. Footnote for how FCR was calculated is missing.

###FCR, kg feed consumed per 12 eggs produced.

####Weight gain over period of study.\\

Page 9:

Line 12: Fermentation? After which fermentation? Clarify?

Line 17: Reference is missing.

 It is known that after fermentation of eggshells to produce alkaline protease, the protein hydrolyzate of egg shells contains several essential amino acids including arginine, isoleucine, leucine, lysine, methionine, phenylalanine, and valine along with smaller quantities of cysteine; thus, eggshells have been proposed as a viable protein source (Nagamalli et al., 2017). 

Line 19: Incomplete reference. Year is missing in reference?

As well, changes in epimerization of isoleucine in eggshell have been used to date the range of age for archeological sites (Brooks et al., 1990). 

Page 14:

Line 12: Were differences within an acceptable range? What is a reduction in levels indicating? Good or bad?

Harlap et al. (2021) studied blood proteins in laying hens (Lohmann Whites at 26- to 80-weeks-old). They reported that the quantity of globulins was not related to the productive period but rather depended on a reduction in defenses with the greatest decrease at 80 weeks of age. At 30 weeks old, our control layers had globulin levels at or above 0.81g/dL (0.62% isoleucine to 0.75% isoleucine) that were gradually reduced to 0.35 g/dL at 0.84 % isoleucine. The reduction of globulins indicated that increasing the isoleucine level beyond 0.75 may be related to a reduction of immunological defenses of the layers. This finding may be important; however, it is not clear if findings for Lohmann Whites are similar to all other breeds of layers, including LSL-LITE layers. Clearly, more research on these findings is indicated.

Line19: which viral strain?

The genetic variation or subtype, susceptibility of host, age, breed, and environment (such as environmentally controlled houses versus open houses) are factors likely affecting ND titers of layers in our work and that of others (Oberlander et al., 2020). 

Page 15:

Rewrite conclusion section: 

Conclusion

 Data from this study is useful because researchers and producers can conduct regression analyses to determine the optimum level of isoleucine for various parameters. For the purpose of our work, it was important to determine a specific level of isoleucine as an important parameter to compare to other BCAA's. In the poultry industry, FCR is most often selected as a important useful performance variable in determining the most appropriate diet (Best, 2011; Clark et al. 2019). In our work, isoleucine levels at 0.72% and 0.84% produced low FCR of 1.45 and 1.44, respectively, but shared significance of FCR (1.49) with 0.78%. Also, at 0.72% isoleucine, other egg production, egg quality, and biochemical measurements were in the medium to high range. Considering cost of synthetic isoleucine as a dietary ingredient along with the low FCR, 0.72% was chosen as the constant for future studies for comparison of valine and/or leucine levels in layer diets. for further study to . 

Line 4 and 5: Delete the sentence. This is not your conclusion. Line 8: Remove reference from the conclusion section. The authors can use this reference in the discussion section.

Line 11? Further studies to test what? Would you please clarify which “further” studies are required? To investigate what?

See above in Conclusion.

Page 15 vs 16: Material and Methods: Line 14 on page 15 says you had 10 layers per replicate, but in Line 13 on Page 16, authors say there is five birds/cage? Please check and confirm if you had 5 or 10 laying hens/replicate?

Two adjoining cages (60 x 63.5 cm, five birds each) constituted a replicate. 

Page 17:

Line 1: Delete the line “ except for layers …..below.” Culls are not counted as Mortality.

There was no mortality. except for layers used to determine ileal digestibility as described below. 

Line 13: what was the level of celite used?

Celite (2 % of diet) was added three days before the end of the study.

Line 20: Revise sentence. 

Remove the word “above” and mention that it was ileal digesta.

After blood samples were obtained (above), cervical dislocation was performed to obtain ileal digesta (González Alvarado et al., 2007; Walk et al., 2012; Xu et al., 2015; and Cunniff, 2016).

Reviewer #2: (No Response)

---

## [Editor Report · Decision Letter 2]

26 Nov 2021

Varying isoleucine level to determine effects on performance, egg quality, serum biochemistry, and ileal protein digestibility in diets of young laying hens

PONE-D-20-35024R2

Dear Dr. Annie King,

We’re pleased to inform you that your manuscript has been judged scientifically suitable for publication and will be formally accepted for publication once it meets all outstanding technical requirements.

Kind regards,

Mahmoud A.O. Dawood, PhD

Academic Editor

PLOS ONE
---

## [Editor Report · Acceptance letter]

19 Dec 2021

PONE-D-20-35024R2 

Varying isoleucine level to determine effects on performance, egg quality, serum biochemistry, and ileal protein digestibility in diets of young laying hens 

Dear Dr. King:

I'm pleased to inform you that your manuscript has been deemed suitable for publication in PLOS ONE. Congratulations! Your manuscript is now with our production department. 

Kind regards, 

on behalf of

Dr. Mahmoud A.O. Dawood 

Academic Editor

PLOS ONE